# Airbnb Hospitality: Exploring Users and Non-Users' Perceptions and Intentions

**Asad Mohsin** [1,*] **and Jorge Lengler** [2]

[1]  School of Management and Marketing, University of Waikato, Hamilton 3240, New Zealand
[2]  Department of Marketing and Management, Business School, Durham University, Durham DH1 3LB, UK; jorge.lengler@durham.ac.uk
[*]  Correspondence: asad.mohsin@waikato.ac.nz

**Abstract:** Although the use of Airbnb services is growing, research relating to its value, risk, satisfaction, and repurchase intentions involving the millennial generation is scarce. This study investigates actual experience of social, utilitarian and hedonic values, risks, satisfaction, and repurchase intentions of Airbnb millennial consumers. It further assesses perceptions of similar values, risks, and what it would take to generate satisfaction and repurchase intention amid Airbnb non-consumers. PLS Path Modelling is used to test the hypothesised relationships and compare Airbnb consumers and non-consumers. A conceptual model proposing five hypotheses is tested using a dataset of 206 responses representing consumers and non-consumers from New Zealand. The results of the two groups are compared to assess differences in the relationships specified in the proposed conceptual model. The findings have theoretical, managerial, and social implications as it expands the literature by comparing consumers/non-consumers relating to sharing economy and identifies factors that lead to satisfaction and repurchase intentions linked to the millennials, hence generating managerial implications. The findings also suggest social, utilitarian, and hedonic values that have implications for the millennial generation.

**Keywords:** sharing economy; Airbnb; values; risk; satisfaction; repurchase intention; millennials

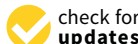



## 1. Introduction

The Airbnb initiated in 2008 in San Francisco (USA) as a peer-to-peer platform has taken commercial accommodation to new heights of innovative experiences, posing new challenges for the traditional hotel/motel industry. Though usage of Airbnb was limited around 2011, it has since grown at an astonishing pace [1–3]. In the sharing economy, peers work as service providers (e.g., hosts of Airbnb and Uber drivers) for consumers (e.g., guests using Airbnb and passengers using Uber). While peers make the transaction directly, the company (e.g., Airbnb and Uber) functions as a platform provider [4]. PWC [5] predicted that by 2025 the sharing economy would reach $335 billion, among which peer-to-peer accommodation becoming one of the most significant growth sectors reaching 5.6 million listings as of 30 September 2020. Different from the traditional market, the sharing economy not only brings economic profits, but also plays a significant role in providing environmental and social benefits [6–8]. Airbnb is described as a business platform where customers can begin by searching for accommodation based on destinations, travelling time, price, and size. In addition, Airbnb helps with listings based on the neighbourhoods, amenities, additional descriptions, photos, and reviews by previous guests [1].

Airbnb [1] indicates that the growth of Airbnb is linked to several factors, but most noticeable is the huge popularity among the millennial generation consumers, aged between 26 and 40 years old. Millennials are expected to account for two-thirds of all customers and travellers by 2025, and over 80% of millennials demand unique travelling experience that satisfies the hedonic values as their key motive. Mittendorf, Berente, and Holten [9] focus squarely on understanding the behaviours of millennials—the generation important to the

sharing economy. Despite the growth and attention in recent years involving the sharing economy globally [10–12], research originating from different geographical regions on consumers valuation and attributes towards Airbnb is scarce. Recently, authors So et al. [13], in their study about motivations and constraints of Airbnb consumers, suggest that as Airbnb is still a relatively new form of accommodation with no clear operational standards, consumers opinions will vary by groups. This study adds to the literature millennial groups' values, satisfaction, and repurchase intentions and compares it with perceptions of those who have never used the Airbnb service.

Several studies have investigated online consumer behaviour, satisfaction, and trust and its impact on repurchase intention [14–16]. Airbnb has emerged as a market-leading commercial accommodation platform in the space of sharing economy [17]. As the popularity of Airbnb grew in recent times, so too has the research linked to Airbnb service quality, satisfaction, trust, repurchase intention, perceived value, behavioural intention, loyalty, impact on tourism, and impact on hotel accommodation [17–32]. These studies have also emphasised motivation of the hosts and consumers and their hedonic and utilitarian values, satisfaction levels, and repurchase intention. Airbnb is seen to be different from the traditional hospitality/hotel industry, and, hence, perceived risks linked to uncertainty, being able to get in the house, negative consequences of booking using Airbnb, security and safety have generated interest [6,8,33].

Nonetheless, research on the impact of personal values on millennial consumers of Airbnb remains limited and a comparison between experienced and inexperienced consumers of the Airbnb is almost non-existent [34–36]. Airbnb listings vary in many ways, hence, the practical and experimental benefits they offer may not always go hand in hand [36]. So, what are the opinions of the millennial generation about Airbnb in both cases, that is, whether they have used or not used the platform? To answer the question, the main objectives of the study are to:

- Gauge the impact of social, utilitarian, and hedonic values on satisfaction with the use of Airbnb services across two millennial groups-consumers and non-consumers.
- Examine the differences on the risk perception and satisfaction with Airbnb service across the two groups of consumers.
- Gauge the relationship between satisfaction and repurchase intention of Airbnb services across the two groups of consumers.
- Assess whether the Airbnb consumers and non-consumers have similarities in their behaviour towards the relationships stated above.

Besides, risk perceptions could be different between the Airbnb consumers and the inexperienced non-consumers. Thus, this study responds to the calls that more research in the sharing economy area is needed to clarify the effects of millennials' values and perceptions linked to Airbnb services [13].The study discusses the outcomes of a newly developed conceptual model and six hypotheses, linked with the two Airbnb groups (see Figure 1). To test hypotheses PLS (partial least squares), path modelling is used because of its flexibility in handling small sample sizes [37].

## 2. Literature Review

### 2.1. Sharing Economy-Airbnb

A quick Google Scholar search reveals that the majority of studies linked to Airbnb were published in the last decade. Most of these studies originated from USA, Canada and Europe, and Asia has in recent years has attracted some attention [38]. The sharing economy has been keenly researched. Bostsman and Roger [4] discussed the pros and cons of this phenomenon, Cusumano [39] and Denning [40] studied the growing threats and challenges, Botsman and Rogers [4] classified how the sharing economy can be analysed, such as the environment that encourages the sharing economy functioning and its implications. Aviatal et al. [41] and Bucher et al. [6] emphasised sharing platform functions in terms of human interaction. Hamari, Sjöklint, and Ukkonen [42] focused on the social perspective. They studied the motivation of sharing and its influence on society and

regulations. The emergence and expansion of Airbnb within a decade of its inception has generated enormous attention as one of the most innovative business ventures of recent decades with the results noted above [43]. Though the 2020 pandemic impacted demand as with other commercial accommodation, the expansion of Airbnb is evident from the fact that from a very slow start in 2008, it grew to over five million active worldwide listings in 2018 [44]. Guttentag [12,25] provides a literature review on the progress of Airbnb and identifies six themes, e.g., Airbnb guests, Airbnb hosts, Airbnb supply and its impact on destinations, Airbnb regulations, Airbnb's impact on the tourism sector, and the Airbnb Company.

Although Tussyadiah [45] studied consumer satisfaction within Airbnb, limited attention has been given to the post-purchase behaviour of Airbnb consumers. Some researchers have studied motivation and satisfaction of the consumers of Airbnb, with a limited focus on repurchase intentions linked to the millennial generation. Others have focussed on perceived values as a homogenous group [19]. Considering that Airbnb listings vary in many ways, such variations may have implications for consumer repurchase intentions. Barnes and Mattson [46] highlighted the economic factor as the leading driver, followed by technology familiarity, socio-cultural benefits, and environmental concerns (i.e., sustainability). Similarly, Hamari et al. [42] confirm that sustainability, enjoyment, and economic gains are the main factors that encourage people to participate in a sharing economy.

### 2.2. Airbnb's Consumers and Non-Consumers

Research exploring the differences between Airbnb consumers and non-consumers' is of value to help uncover factors thought important for marketing and the growth of Airbnb. A study by Poon and Huang [47] compared Airbnb consumers and non-consumers' perception. The study found that both consumers and non-consumers expressed few differences in their perceived importance of accommodation attributes, but the two groups vary in their perceptions and evaluation of Airbnb when compared to hotels. Specifically, they preferred hotels when travelling with family for a shorter trip, whereas Airbnb was preferred when travelling with friends and for longer trips. The current study is very different as it assesses the impact of social, utilitarian, and hedonic values of the millennial generation consumers and non-consumers. It also assesses risk perceptions and repurchase intentions.

Recent studies have explored regulatory issues about Airbnb [48–50], discrimination [51,52], Airbnb branding strategies [53,54] and its influence on the hotel industry [35,55]. Airbnb also provides a platform for individuals to participate in hospitality entrepreneurial activities [34] as it has grown as an alternative global accommodation choice [35]. When it comes to the perceived risks, new Airbnb hosts (those with less than six months of hosting experience) are more worried about the security and being taken advantage of by consumers. Ikkala and Lampinen [56] found that to select the right guests, some hosts are lowering their listing prices. This may indicate that the hosts are not only seeking economic benefits but also care for emotional and social values. Guttentag et al. [12,25] have noted that people are more strongly attracted to Airbnb by its practical attributes than by its experiential attributes. These studies inspire to investigate the motives and values of Airbnb consumers.

### 2.3. Theoretical Background and Hypotheses Development

To investigate the perceptions of Airbnb's consumers and non-consumers, and their relationship to the social, utilitarian, and hedonic values, this study reviewed two main theoretical backgrounds: (1) the MEC (means-end chain) by Gutman, [57] and (2) the prospect theory proposed by Kahneman and Tversky [58].

The MEC model by Gutman [57] connects customer value and customer behaviour. It presents customers' cognitive abstractions in three hierarchical levels (i.e., attributes, consequences, and values). Thus, Gutman's model considers values as the main drivers of customer behaviour, and customers' choice patterns. Therefore, the MEC model is

commonly used as a theoretical framework for investigating the differences between motivations and perceived values [59]. An empirical study of customer value perception is considered essential in the tourism marketing area. Particularly, the means-end model has been widely adopted in investigating the perceived values [60]. Perceived value is defined as the consumer's assessment of the benefits of the services that lead to satisfaction [61,62].

Lawson et al. [63] suggest that consumers who frequent Airbnb accommodations consider they are pushing a prosocial movement and that social values are important determinants of their satisfaction. Certainly, Guttentag et al. [12] found that social value is an important driver based on the consumers' increasing awareness of a perceived sustainability in sharing consumption that is related to social or altruistic values. Hence, Airbnb is perceived to be a social and sustainable value-oriented service, which helps in reducing environmental harms, recycling resources, and purchasing products not associated with mass commercial activity. For example, first, it can help to reduce building more motels and hotels. Second, it can also support local residents and the local economy [4,63]. It raises a question whether social value has any impact on satisfaction within the context of Airbnb use amongst millennials and whether it differs between consumers and non-consumers. Thus, our H1 states:

**Hypotheses 1 (H1).** *Social values will positively affect the satisfaction with the Airbnb experience.*

The second construct assesses utilitarian values' relationship and impact on satisfaction in using the Airbnb services. Motivations that drive an action can be categorised as extrinsic and intrinsic [64]. Extrinsic motivation relates to the performance of an activity, which leads to a utilitarian benefit and thereby utilitarian value. Several authors have identified convenience, broad product offerings, rich product information, and monetary savings as benefits with utilitarian values in online shopping [31,65–69]. The utilitarian values are identified as one of the main drivers of a sharing economy related to accommodation [4,35,70–73]. Our hypothesis H2 explores the effects of utilitarian values on the satisfaction with Airbnb services:

**Hypotheses 2 (H2).** *Utilitarian values will positively affect the satisfaction with the Airbnb experience.*

The third value being assessed in this study relates to the hedonic benefits. These benefits include adventure, social, enjoyment, and sense of belonging. That Airbnb can create and maintain social bonds can be interpreted as the enjoyment of collaborative consumption [4,35,74–76]. Many Airbnb listed accommodations offer hedonic values by being different, unique, and with an authentic home experience [30]. Customers can also fulfil their hedonic values from their prosocial behaviours [77–79]. Thus, our hypothesis H3 explores the effects of hedonic values on the satisfaction among Airbnb consumers and non-consumers:

**Hypotheses 3 (H3).** *Hedonic values will positively affect the satisfaction with the Airbnb experience.*

Kahneman and Tversky [58] developed prospect theory to describe the process of an individual's decision-making under uncertainty and risk. The study by Meng and Weng [80] further pointed out that an individual's behaviour is based on two aspects, namely the outcome of the action (i.e., loss or gains) and risk attitudes. In general, people are loss-averse, and they perceive loss as a more significant and fundamental impact than getting an equal amount of gains psychologically speaking [58]. In this case, the perceived risks stand for the potential loss while the perceived value means the gains after comparing the benefits and sacrifices received when pursuing a desired outcome [81]. Prospect Theory has been widely used in modelling and predicting human behaviour [82]. Ponte, Carvajal-Trujillo, and Escobar-Rodríguez [83] suggest that perceived value has a positive influence on customer behaviour while perceived risks have a negative impact on customer behaviour.

The research in relation to risk perception linked to Airbnb is scarce [50]. A study by Liang [19] points out that perceived risk in Airbnb is present amongst consumers expecting

several negative consequences after booking the Airbnb accommodation. Similarly, the Airbnb hosts also have several negative expectations related to the guests being unknown. In assessing perceived risk specific to Airbnb, Liang et al. [19] found that perceived authenticity was negatively correlated with perceived risk. Do Airbnb consumers often fear whether they will be able to get into the house? Will they be secure? What might be the quality of the accommodation? Forsythe et al. [84] have noted such factors as perceived risks. Those factors could influence the satisfaction specifically in the case of inexperienced potential consumers. On the other hand, most of these factors are entirely secure when one books a quality commercial hotel accommodation and perceived risks are almost non-existent. Consumers who have the intention to use Airbnb are willing to concentrate on seeking the 'right' accommodation despite the risk perceptions. Hence, how does perceived risk impact satisfaction in the case of Airbnb consumers? Thus, our H4 tests whether the perceived risk has a negative effect on satisfaction with Airbnb services:

**Hypotheses 4 (H4).** *Perceived risk will negatively affect the satisfaction with the Airbnb experience.*

Park and Kim [85] claimed that when continuous satisfaction is generated amongst consumers, brand loyalty kicks in. Hence, satisfaction with the use of Airbnb will happen when customers perceive Airbnb services as providing value and meeting the needs and criteria of all consumers and hosts. Satisfaction is defined as the confidence in the reliability of a company to significantly and positively influence customer behaviour [86]. Studies by Chou, Chen, and Lin [87] and Ku [88] highlighted that satisfaction can generate trust through the initial as well as the customer retention stage. Further, satisfaction can also have a positive impact on customer loyalty [89]. Chitty, Ward, and Chua [90] defined customer satisfaction as the comparison between the sacrifices (i.e., costs) and rewards (i.e., benefits) during the consumption process. In the hospitality industry, satisfaction is defined as the extent to which the hotel stay meets the customer's desires, expectations, and requirements [91,92]. Yee et al. [93] find that offering more relational benefits positively impact satisfaction and repurchase intentions in internet banking. The impact of satisfaction has also been noted by Jeong, Oh, and Gregoire, [94] highlighting that, the higher the level of satisfaction, the greater the chance for customers to visit the destination again. As a result, it could also be expected that the positive effect of satisfaction on repurchase intention can be mirrored in the Airbnb service. Thus, hypothesis H5 below:

**Hypotheses 5 (H5).** *Satisfaction with the Airbnb experience will positively affect the repurchase intention of the service.*

Prospect theory [58] suggests that people's evaluation of alternatives available to them impacts their behaviour, their evaluations are linked to outcomes or expected outcomes and their risk attitudes. Usually based on outcomes and its associated probabilities, people choose the alternative with the highest value. Prospect theory has been applied in modelling and predicting consumer behaviour [81]. How does it affect commercial accommodation alternatives? To expand the literature by studying the perceptions of consumers who have not used the Airbnb services with those who have, H6 is proposed. The hypothesis tests the relationships between perceived social, utilitarian, and hedonic values, perceived risk, and satisfactions of non-consumers with actual experience of consumers.

**Hypotheses 6 (H6).** *Consumers and non-consumers will have similar behaviour regarding the relationships of the model.*

Hence, based on the above six hypotheses, a new conceptual model is proposed to test the relationships between social, utilitarian, and hedonic values, perceived risk, and satisfaction within the context of Airbnb and its millennial consumers. The model also tests the impact of satisfaction on repurchase intention and if there are differences between consumers and non-consumers of the Airbnb (see Figure 1). The model and hypotheses draw upon a literature review [4,18,19,25,35,63,65–68,73,75–77,80,82,89,95] and adapts to the objectives of the current study. The study is expected to contribute some new insights through the test results of the conceptual model proposed and how these relationships

compare with non-consumers of Airbnb. An overview of the conceptual framework is presented in Figure 1 below.

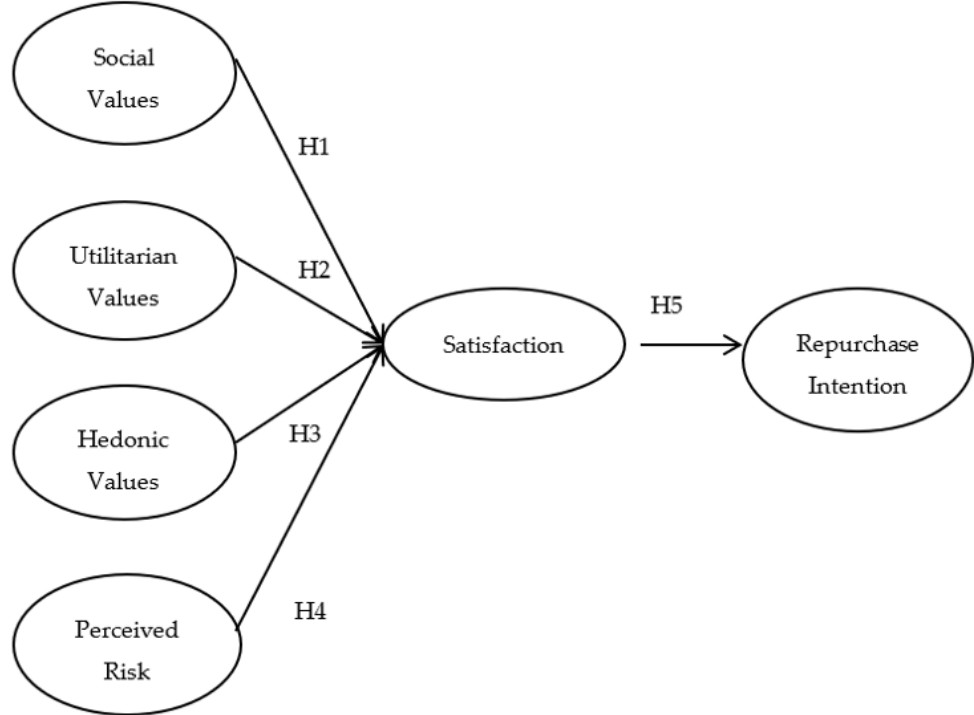

**Figure 1.** Conceptual Model.

### 3. Methodology

*3.1. PLS Path Modelling and Multi-Group Analysis*

Analysis is done using PLS (partial least squares) path modelling in this study because of its flexibility in handling small sample sizes [37,96]. There are 103 cases in each subsample. Authors initially estimated the proposed model (Figure 1) using an aggregated dataset combining Airbnb consumers and non-consumers. In the second stage, we tested the effects of the different subsample groups (consumers and non-consumers) in the hypothesised relationships. To do so, we included a discrete moderator variable, which is interpreted as dividing our aggregated dataset into two subsamples [96,97]. The moderator used in our model is the user category (consumers and non-consumers). Thus, as it is not possible to compare groups is PLS using a global criterion, we compared the path coefficients of the subsamples [96]. By doing this, it enabled us to assess path coefficient differences between the two groups.

According to Chin [97], the path coefficient significance differences across subsamples can be tested with pairwise *t*-tests. To do so, we assessed all subsample models' goodness-of-fit. Multigroup analysis was carried out using measurement invariance. The permutation results indicate that there are no differences between the users and non-users measurement models.

*3.2. Sample and Sampling Measures and Procedures*

In order to understand the personal values, perceived risks, satisfaction, and repurchase intentions within the context of Airbnb among millennials, this study uses a survey questionnaire for primary data collection. Typically, research related to consumer behaviour largely uses two sampling methods, namely snowball sampling and purposive sampling [98]. The data was collected in two groups, i.e., consumers who have used Airbnb and non-consumers who have not used Airbnb. A couple of filter questions were asked to assess the suitability of responses. The first filter question was to determine whether they

represent the millennial age group (born in the period 1980–1994), the second to determine if they have experienced using Airbnb service to qualify as 'users'. If respondents had not used Airbnb, they were segmented as 'non-users'. The respondents who had used Airbnb shared their actual experiences relating to values, risks, satisfaction, and repurchase intention. The non-users with an intention to use Airbnb services in the future were encouraged to share their perceptions relating to the values and risks, assign a value to each measure, and indicate if these would impact their perceived satisfaction. Moreover, they were asked if their perceived satisfaction would generate repurchase intention of the service. All respondents represented the millennial generation born between 1980 and 1994 (26–40 years) in New Zealand. Appendix A presents sample characteristics. A snowball sampling method was used in this case. Social media and personal networking were used to accumulate responses. The total sample size consisting of 206 observations was used to test relationships as stated in Figure 1. A comparison test was also done to assess differences/similarities of the two subsamples: Airbnb consumers (*n* = 103, 50%) and non-consumers (*n* = 103, 50%). All survey items are evaluated using a 7 Point Likert Scale, ranging from "1 = strongly disagree" to "7 = strongly agree". The measures used in the constructs and their sources are identified in Table 1. The impact of values and perceived risk on satisfaction and impact of satisfaction on repurchase intention is tested using hypotheses 1 to 5. Similarities/differences in the behaviour between the Airbnb consumers and non-consumers are tested through hypothesis 6.

**Table 1.** Latent Variables Composition and Variable Loadings [12,25,30,79,99–102].

| Latent Variables | Definition | Measures | Loading | Adapted from |
|---|---|---|---|---|
| Utilitarian Value | The definition of utilitarian value is the value that comes from the task related and rational consumption (Babin et al., 1994) [79]. | Money saving | 0.858 | (Homer and Kahle's, 1988) [99] |
| | | A wide variety of housing | 0.806 | |
| | | Ease to use | 0.732 | |
| Hedonic Value | The definition of hedonic value is the value that a customer received from the fun and enjoyable experience of consumption (Babin et al., 1994) [79]. | Sense of belonging | 0.752 | (Homer and Kahle's, 1988) [99] |
| | | Friends making | 0.817 | |
| | | Experience the local culture | 0.834 | |
| Social Value | The definition of the social value here refers to a social and sustainable value, which helps in reducing environmental harms, recycling the resources, and purchasing products not associated with mass commercial activity. | Natural resources saving | 0.923 | (Mohlman, 2015, Guttentag et al., 2017) [12,25] |
| | | Help with local economy | 0.889 | |
| Perceived Risk | Perceived risk is defined as the uncertainty that customers feel when they are consuming a goods or services. | Safety and privacy concerns | 0.895 | (Chai et al., 2011; Bhattacherjee, 2002) [100,101] |
| | | Uncertain house/service quality | 0.920 | |
| Satisfaction | Satisfaction is defined as the degree to which the customer feels satisfied about the Airbnb. | Ideal accommodation | 0.769 | (Fornell et al., 1996) [102] |
| | | Expectation fulfilment | 0.892 | |
| | | Overall Satisfaction | 0.905 | |
| Repurchase intention | The repurchase intention is defined as the likelihood of customers choosing Airbnb again. | I am likely to choose Airbnb next time. | 1.000 | (Bhattacherjee, 2001) [101] |

## 4. Results

### 4.1. Model Estimation and Validity

　　　Table 2 shows the path coefficient results for the relationships estimated in the overall model (aggregated dataset) and each of the subsamples of this study (Figure 1). To estimate the model, we followed the procedures recommended by Sarsted et al. [103] and calculated the model using the bootstrapping procedure with 5000 subsamples. Hence, we used a large number of subsamples on the bootstrapping calculation following PLS' multi-group analysis procedure as this ensures stability of the results [104].

**Table 2.** Overall Sample and Value-Specific Results Analysis.

| | | Overall Sample $n = 206$ | Airbnb Consumers $n = 103$ | Non-Consumers $n = 103$ |
|---|---|---|---|---|
| Latent Variables | | | | |
| Hedonic Values | Composite reliability | 0.843 | 0.841 | 0.817 |
| | AVE | 0.642 | 0.639 | 0.598 |
| | Cronbach's Alpha | 0.720 | 0.715 | 0.663 |
| Social Values | Composite reliability | 0.902 | 0.864 | 0.927 |
| | AVE | 0.821 | 0.761 | 0.864 |
| | Cronbach's Alpha | 0.780 | 0.700 | 0.842 |
| | Spearman Brown Coefficient | 0.784 | | |
| Utilitarian Values | Composite reliability | 0.842 | 0.828 | 0.838 |
| | AVE | 0.641 | 0.618 | 0.638 |
| | Cronbach's Alpha | 0.722 | 0.695 | 0.723 |
| Perception of Risk | Composite reliability | 0.903 | 0.927 | 0.821 |
| | AVE | 0.823 | 0.863 | 0.704 |
| | Cronbach's Alpha | 0.786 | 0.842 | 0.682 |
| | Spearman Brown Coefficient | 0.786 | | |
| Satisfaction | Composite reliability | 0.892 | 0.885 | 0.895 |
| | AVE | 0.735 | 0.722 | 0.740 |
| | Cronbach's Alpha | 0.822 | 0.806 | 0.827 |
| Repurchase intention | Composite reliability | 1.000 | 1.000 | 1.000 |
| | AVE | 1.000 | 1.000 | 1.000 |
| | Cronbach's Alpha | 1.000 | 1.000 | 1.000 |
| Relationships | | Coefficient and significance level | | |
| H1: Social Values → Satisfaction | | 0.288 *** | 0.259 *** | 0.372 *** |
| H2: Utilitarian Values → Satisfaction | | 0.066 | 0.139 ** | 0.011 |
| H3: Hedonic Values → Satisfaction | | 0.362 *** | 0.454 *** | 0.203 ** |
| H4: Perception of Risk → Satisfaction | | −0.118 *** | −0.159 * | −0.065 |
| H5: Satisfaction → Repurchase intention | | 0.654 *** | 0.728 *** | 0.567 *** |
| R2 Satisfaction | | 0.419 | 0.583 | 0.281 |
| R2 Repurchase Intention | | 0.428 | 0.530 | 0.322 |
| Standardised Root Mean Square Residual (SRMR) goodness-of-fit assessment (GoF) | | 0.035 | 0.054 | 0.050 |

Notes: AVE = Average Variance Extracted; * Significance at 0.10, ** significance at 0.05, *** significance at 0.01.

　　　According to Chin [97], the model and data to compare multiple groups with pairwise *t*-test have to satisfy three assumptions: (1) the data should not be too non-normal; (2) each sub model has to present acceptable goodness-of-fit, and (3) the measurement across the different sub models must be invariant. The first condition has been verified through an analysis of Skewness and Kurtosis results. The Skewness results for all variables in the measure model fall within the range of −1 to +1 defined by the literature [105]. Regarding the Kurtosis, only one variable (Sense of belonging) displayed result larger than

2, allowing us to consider that thirteen variables have not deviated from the distributional assumption [96,97].

The second condition has been assessed by different criteria. Authors assessed content validity based on the literature review and by consulting experts in the area of tourism management. Based on these initial procedures, it was concluded that the measures used in this study had content validity. Our analyses also reveal that all measures displayed loadings above the cut-off point of 0.70 suggested by the literature [106,107] (Table 1). Regarding the reliability of the constructs, our results indicate that all average variance extracted (AVE) values were above 0.50 (Table 2). Table 2 also shows that all constructs' composite reliability values were higher than 0.80. Cronbach's Alpha ($\alpha$) has been calculated for all constructs on the overall sample and the two subsamples. The results of Cronbach's alpha indicate that all constructs on the three samples were internally consistent ($\alpha \geq 0.60$; [105]). However, Eisinga, Grotenhuis, and Pelzer [108] advocate that Cronbach's alpha is not an adequate reliability measure for two-item factors. As two of our constructs (Social Values and Perceived Risk) were based on two observed variables each, we followed the procedure proposed by Eisinga et al. [108] calculated the Spearman–Brown coefficient for those two factors. According to the authors, the Spearman–Brown coefficient is the most appropriate reliability statistic for a two-item scale. The results are shown in Table 2. Authors also assessed the discriminant validity of all constructs. This was assessed based on two measures. The first was the analysis of Fornell and Larcker's [109] approach to assess the construct inter-correlations. Table 3 indicates that construct correlations were significantly different from 1 and the shared variance between any two constructs in the model was less than the average variance explained in the items by the construct. Due to the low sensitivity of traditional methods (i.e., [109] method) in assessing discriminant validity in variance-based structural equational modelling, we have applied Henseler, Ringle, and Sarsted [110]'s Heterotrait–Monotrait Ratio (HTMT) of the correlations. Henseler et al. [110] advocate that HTMT provides more accurate results for establishing discriminant validity in PLS. The HTMT is the average of the heterotrait-heteromethod correlations relative to the average of monotrait–heterotrait correlations of the constructs [110]. The results of our calculation (Table 4) indicate that the HTMT values for any pairs of constructs in the model are lower than the HTMT threshold of 0.85 established by Kline [111]. These results indicate that the constructs have discriminant validity. Further, we checked the whole model and each submodel's R2 in regards to the two endogenous in each subsample. Table 2 indicates that the R2 values for the endogenous constructs satisfaction with Airbnb service and repurchase intention have been deemed acceptable for the aggregate sample model and each of the subsample models. The R2 values obtained are within the range of results obtained by other studies in the area e.g., [112] and above the cut-off point of 10% defined by Falk and Miller [113]. In addition to the R2 results, we also assessed the overall model's goodness-of-fit based on Hu and Bentler's [114] standardised root mean square residuals (SRMR). Henseler et al. [103] propose that the SRMR as a goodness-of-fit measure can avoid model misspecification. The SRMR is the difference between the observed correlation and the model implied correlation matrix. As a result, SRMR allows assessing the average magnitude of the discrepancies between observed and expected correlations as an absolute measure of fit criteria. According to Hu and Bentler [114], SRMR values lower than 0.08 indicate good fit. The overall model, consumers sample model, and non-consumers sample model displayed SRMR values of 0.035, 0.054, and 0.050, respectively (Table 2). Thus, the three models are deemed acceptable.

Finally, we assessed the invariance of the six constructs across our two subsamples. The measurement invariance is assessed by checking the loadings and weights of our indicators to each construct across the different subsamples. The results of the variance of the measurement model checks indicate that all six constructs are invariant ($p > 0.10$) across the two subsamples.

**Table 3.** Correlation Matrix of Constructs.

| Latent Variables | Hedonic Values | Perceived Risk | Repurchase Intention | Satisfaction | Social Values | Utilitarian Values |
|---|---|---|---|---|---|---|
| Hedonic Values | **0.801** | | | | | |
| Perceived Risk | −0.239 | **0.907** | | | | |
| Repurchase Intention | 0.656 | −0.157 | **1.000** | | | |
| Satisfaction | 0.574 | −0.274 | 0.654 | **0.858** | | |
| Social Values | 0.520 | −0.188 | 0.613 | 0.527 | **0.906** | |
| Utilitarian Values | 0.523 | −0.234 | 0.508 | 0.407 | 0.428 | **0.800** |

Note: the diagonal is the square root of the Average Variance Extracted (AVE) of the constructs in the whole sample.

**Table 4.** Heterotrait–Monotrait Ratio of Correlations (HTMT) Results.

| Latent Variables | Hedonic Values | Perceived Risk | Repurchase Intention | Satisfaction | Social Values |
|---|---|---|---|---|---|
| Hedonic Values | | | | | |
| Perceived Risk | 0.315 | | | | |
| Repurchase Intention | 0.769 | 0.176 | | | |
| Satisfaction | 0.726 | 0.325 | 0.699 | | |
| Social Values | 0.686 | 0.231 | 0.690 | 0.640 | |
| Utilitarian Values | 0.719 | 0.313 | 0.588 | 0.495 | 0.563 |

The following section discusses the results of the hypotheses testing on the overall model (aggregated dataset) and the moderating effects of each of the subsamples.

*4.2. Structural Results*

The analysis tested the proposed model and the relationships in the overall sample and two subsamples. The results of the model estimation and the goodness-of-fit have been displayed in Table 2. The R2 results of the dependent constructs repurchase intention of Airbnb services and satisfaction with Airbnb services have been deemed acceptable in the three models (aggregate and two subsamples) estimated in this study.

The results of the overall sample (aggregate) model path coefficients indicate that social values (0.288; $p < 0.001$) have a strong positive effect on satisfaction with Airbnb service. This result provides support for H1.

Our hypothesis H2 proposed that utilitarian values have a positive effect on the satisfaction with the Airbnb experience across the overall sample. The results obtained for the path coefficient fail to provide support for H2 regarding the overall sample (0.066; $p > 0.10$).

However, our results support the notion that hedonic values positively affect the satisfaction with Airbnb services (0.362; $p < 0.001$) in the overall sample. This result provides support for H3.

Perceived risk of using Airbnb, in its turn, has negatively affected satisfaction with the service in the overall sample (−0.118; $p < 0.001$), providing support for H4. The path coefficient results also suggest that the satisfaction with the Airbnb service has a strong positive effect on repurchase intention among the aggregate sample (0.654; $p < 0.001$). Thus, H5 has been accepted.

To test if the effects of personal values and perceived risk on satisfaction with Airbnb service and satisfaction on repurchase intention would statistically differ between the two subsamples of the study (consumers and non-consumers), we tested the path coefficient differences of the estimated relationships of the proposed model (Figure 1). This this analysis is described in the next section.

Assessing the Differences between Airbnb Consumers and Non-Consumers

The results of the bootstrapping procedure indicate that social values positively influenced satisfaction with Airbnb services in the two subsamples of our study (consumers: 0.259; $p < 0.001$, and non-consumers: 0.372; $p < 0.001$). In regard to the effect of utilitarian values on satisfaction, it can be concluded that the result was positive and significant in the consumers' sample (0.139; $p < 0.05$). However, in the case of non-consumers, the effect of utilitarian values on perceived satisfaction with use of Airbnb activities was not significant (0.011; $p > 0.10$).

Our multi-group model estimation indicates that hedonic values positively influenced satisfaction in the two samples (consumers: 0.454, $p < 0.001$; non-consumers: 0.203; $p < 0.01$).

Perceived risk negatively affected satisfaction with Airbnb services in one of our subsamples (consumers: $-0.159$, $p < 0.05$). The relationship between perceived risk and perceived satisfaction with Airbnb services was not significant for the non-consumers subsample ($-0.065$, $p > 0.10$).

The last relationship tested in our model across the two subsamples was the influence of satisfaction with Airbnb service on the repurchase intention. The path coefficients of those relationships were positive and significant in both subsamples, indicating that for consumers (0.728, $p < 0.001$) and non-consumers (0.567, $p < 0.001$) the satisfaction with Airbnb service did positively affect their repurchase intention.

Despite the numeric differences verified between the two subsamples regarding the path coefficients of each hypothesised relationship, we assessed if those differences were statistically significant. This procedure allowed us to test research hypotheses H6. Authors assessed those differences by comparing path coefficients of the two subsamples [110,115]. Table 5 displays the numeric difference within pairs of subsamples (|diff|), the $t$-values (parametric), and the $t$-values after permutation. Although the numeric differences are large between some of the path coefficients of the two subsamples, only in one case has the difference been considered statistically significant (hedonic values–satisfaction relationship). Ahead is the analysis of those path coefficient differences and H6 testing.

**Table 5.** Multi-group Comparison Results–H6 Testing.

| Relationship | Pair Test | \|diff\| | $t_{Parametric}$ | $t_{Permutation}$ |
|---|---|---|---|---|
| Social Values → Satisfaction | Consumers vs. non-consumers | −0.113 | 1.068 | 1.080 |
| Utilitarian Values → Satisfaction | Consumers vs. non-consumers | 0.127 | 1.039 | 1.037 |
| Hedonic Values → Satisfaction | Consumers vs. non-consumers | 0.250 | 2.066 ** | 2.085 ** |
| Perception of Risk → Satisfaction | Consumers vs. non-consumers | 0.094 | 0.721 | 0.722 |
| Satisfaction → Repurchase intention | Consumers vs. non-consumers | 0.161 | 1.681 | 1.670 |

Notes: |diff| = Path Coefficient difference; ** Significance at 0.05.

As the results of Table 5 indicate, the comparison been consumers and non-consumers have not been statistically significant for the social values–satisfaction relationship. This means that consumers and non-consumers have similar behaviour regarding the influence of social values on satisfaction with Airbnb services. Taking into consideration the results presented in Tables 2 and 5, it can be argued that the two samples indicated that social values strongly positively influence satisfaction with Airbnb service.

The results indicate that the difference between consumers and non-consumers' path coefficients to the utilitarian values–satisfaction with Airbnb service relationship was not statistically significant (|diff| = 0.127, $p > 0.10$). Although the path coefficient of that relationship was positive and significant in the consumers' models and non-significant in the non-consumers' model (Table 2), the above results show that that difference was not statistically significant.

The results indicate that the difference between consumers and non-consumers' path coefficients to the utilitarian values–satisfaction with Airbnb service relationship was not statistically significant (|diff| = 0.127, $p > 0.10$). Although the path coefficient of that

relationship was positive and significant in the consumers' models and non-significant in the non-consumers' model (Table 2), the above results show that that difference was not statistically significant.

Consumers and non-consumers presented statistically significantly different path coefficient results regarding the effects of hedonic values on satisfaction. The difference between the two subsamples was 0.250 ($p < 0.05$) (Table 5). These results suggest that the impact of hedonic values on satisfaction is positive and significant for consumers and non-consumers. However, according to Table 5, the impact of hedonic values on satisfaction is significantly stronger in the case of consumers (0.454; $p < 0.001$).

In regard to the relationship between perceived risk of using Airbnb and satisfaction with the sharing economy service, pairwise test results in Table 5 indicate that the path coefficients of the consumers and non-consumers' models were not significantly different ($|\text{diff}| = 0.094$; $p > 0.10$). Although significant and negative for consumers, the path coefficient is very weak ($-0.159$, $p < 0.10$) while the results show that for non-consumers that relationship is not significant.

Finally, we assessed the differences between satisfaction and repurchase intention relationships between the two subsamples. According to the results in Table 5, the pairwise comparisons did not indicate statistically significant differences between consumers and non-consumers regarding the influence of satisfaction with the Airbnb and repurchase intention ($|\text{diff}| = 0.161$; $p > 0.10$). Results from Tables 2 and 5 suggest that the two subsamples displayed similar behaviour regarding the effects of satisfaction with Airbnb service and repurchase intention. Besides, the fact that the path coefficients of that relationship are positive, strong, and significant in the two subsample models (Table 2) indicates that satisfaction with the Airbnb service leads to a repurchase intention regardless the group of stakeholders (consumers and non-consumers).

The multi-group comparison results allow us to partially accept H6. Thus, it can be argued that consumers and non-consumers display similar behaviour in four of the relationships tested in the model. The two groups presented different results regarding the hedonic values and satisfaction with Airbnb services relationship.

## 5. Conclusions and Implications

Airbnb has grown significantly with more than three million listings worldwide [116]. Airbnb is a digital platform enabling transactions between consumers and owners of accommodation [4,42,78,117–119]. Research has grown in the last decade involving the sharing economy and its implications [4,35,70–72]. Recent studies have conceptualized the drivers of the sharing economy, highlighting role of technology [78,120,121], sustainable consumption and production [92], and population growth and urbanization [122]. However, investigations relating to generational impact on values, risks, satisfaction, and repurchase intention of Airbnb services are scarce. Secondly, there is a research gap in knowing and understanding the perceptions about Airbnb of those people who have not used the service. This has been the impetus for the current study. The findings are based on the PLS Path Modelling Multi-Group Analysis of 206 responses, which tested the overall sample and the two subgroups, i.e., consumers and non-consumers involving Airbnb services.

The study sought to test the impact of social, utilitarian, and hedonic values on satisfaction using hypotheses 1 to 3 with the use of Airbnb services by the millennial generation. The overall results of aggregate sample suggest that social values have a strong positive effect on satisfaction, thus, supporting H1. According to Lawson, Gleim, Perren, and Hwang [77], the behaviour of consumers who frequent Airbnb reflects a prosocial movement related to environmental harms, recycling resources, and purchasing products not associated with mass commercial activity. The social values relating to environment and sustainable consumption have relatively matured in the New Zealand society. Customers are widely immersed in environmentally friendly tourism activities. Therefore, it is possible that, comparatively, New Zealanders are more likely to use Airbnb out of the reason of

helping local economies and reducing the waste of natural resources. This notion is supported by the two groups (i.e., consumers and non-consumers) and is in line with the findings of Guttentag et al. [12,25] and Böcker and Meelen [7]. The H2 tested the relationship between utilitarian values and satisfaction, proposing that utilitarian values have a positive effect on satisfaction. The results of the aggregate sample fail to support the notion. This contradicts the findings of Lee and Kim [20,123], which suggest that utilitarian values significantly influence customer satisfaction. Several other studies have also identified utilitarian, economic benefits as key drivers of the sharing economy related to commercial accommodation [4,35,70–72]. Results of the H2 provide an interesting insight as the overall aggregate sample and non-consumers show no statistical significance, but the consumers do show significance. It could be that the non-consumers of Airbnb are not able to appreciate utilitarian values and/or economic benefit of the Airbnb accommodation because they have not experienced it. However, the results in Table 5 show that consumers and non-consumers do not display significantly different results regarding the effects of utilitarian values on satisfaction.

Perhaps the case with the millennial generation is somewhat different. The test results of H3 provide support for the notion that hedonic values positively affect satisfaction in case of the aggregate sample and the two groups of consumers and non-consumers. According to Arnold and Reynold [76], hedonic value includes adventure, social, enjoyment, and sense of belonging. Several studies confirm that Airbnb can create a sense of community, which can be interpreted as the enjoyment of collaborative consumption [4,35,75,95,124].

Perceived risk in using the Airbnb was tested using H4, which proposed that perceived risk will negatively affect satisfaction with the Airbnb services. The results suggest that the notion negatively affected satisfaction in case of aggregate sample and consumers of Airbnb. However, it did not show any significance in case of non-consumers. Ganapati and Reddick [125] stated that "despite its growth in the last decade, the emerging sharing economy is in a flux and fraught with risks" (p. 80). Perceived risk is an important issue intertwined with online transaction platforms that has been considered by many researchers. Malazizi et al. [126] have identified that perception of risk influences satisfaction and repurchase intention. This is supported by the current findings too. Being an online platform, Airbnb does influence behavioural intention linked to risks perceptions [18,19]. The research related to risk perceptions with Airbnb services is scarce. Liang [19] highlights that perceived risk in Airbnb services are present amongst consumers and hosts. However, consumers planning to use Airbnb pay more attention to their own accommodation needs, utilitarian values such as property information, location, and price, despite the perceived risk in making a purchase decision. Such customers usually have low-risk perceptions compared to customers seeking hedonic values [127].

Hypothesis H5 tested the effects of satisfaction on repurchase intention. The relationships were positive and strongly significant across the aggregate sample, consumers and non-consumers. Results confirm the significant positive impact of satisfaction on repurchase intention in previous research [27,90,91,94]. The findings of the current study are in line with these studies that in Airbnb context, the higher the level of satisfaction, the greater the chances of repurchase. The results in the case of non-consumers reaffirm their intention that if they were satisfied with the service they would repurchase.

Lastly, the H6 compared groups of consumer and non-consumer regarding the relationships tested in the Conceptual Model (Figure 1). The results are largely similar but showed difference in the relationship between hedonic values and satisfaction. The non-consumers perhaps have largely experienced staying in a hotel/motel type of accommodation. They may not value adventure, social enjoyment, and collaborative consumption. Perhaps staying in other people's houses impact their experience negatively. These reasons might have led to partial acceptance of the H6.

Overall, the hypotheses test results show how social values, utilitarian values, and hedonic values, risk perceptions, satisfaction, and repurchase intentions are related and how it implicates the aggregate sample and the consumers and non-consumers groups

using Airbnb services. Pairwise comparison has revealed some statistically significant differences among the relationships tested in the model.

*5.1. Theoretical Implications*

The sharing economy and Airbnb are active research topics with its peer-to-peer business model and rapid growth since 2012 [18,19,128]. Some recent studies have investigated main drivers of the sharing economy such as technology [78,120,121,129,130], sustainable consumption and production [92], and population growth and urbanization [122]. The findings of the current study suggest some meaningful and sustainable theoretical implications. First, it expands the literature showing direct effects of social, utilitarian, and hedonic values and perceived risks as antecedents to the satisfaction of the millennial generation consumers and non-consumers of Airbnb services. Secondly, the study reaffirms what previous studies suggest, which is that guests' satisfaction has significant positive impact on repurchase intention of Airbnb accommodation [18,19,90,91,94,131]. However, this contribution expands the literature by linking it to the millennial generation's use of Airbnb. Thirdly, the study also confirms the MEC theory and its effectiveness in showing that perceived values have a positive and significant impact on the satisfaction and repurchase intention. However, in relation to Kahneman and Tversky's [58] prospect theory, which deals with an individual's decision-making under uncertainty and risk, the current study somewhat challenges the theory as it shows that, largely, consumers consider Airbnb as a low risk and focus on getting the right accommodation. Fourthly, in testing the multi-group comparison, the findings suggest that in the overall aggregate model and in case of non-consumers of Airbnb, utilitarian values show no statistical significance in relation to satisfaction, whereas Airbnb consumers show significance. Thus, the study advances the theory by shedding light on the differences between Airbnb consumers and non-consumers. Investigating perceptions of non-consumers of Airbnb adds a new perspective to the current literature.

*5.2. Managerial Implications*

Sharing economy is having a significant impact on employment in the tourism industry [23]. There are several implications that emerge from the current study from which Airbnb consumers and hosts can benefit. The findings suggest that social values generate satisfaction across the two user groups of Airbnb, i.e., experienced consumers and inexperienced non-consumers. Hence, hosts need to consider improving social values by identifying how they can enhance sustainable value, which helps in reducing environmental harms, recycling resources, and purchasing products not associated with mass commercial activity and promote it amongst their consumers, which has immediate implication on sustainability of environment. Woodruff [132] pointed out that it is essential for a company to deliver superior value to the customer in order to stay competitive and succeed in the long term. Malazizi et al. [126] have identified that perception of risk influences eventual satisfaction and repurchase intention and that perceived risks remain an important factor that influences satisfaction rate and in turn repurchase intention. Some possible actions, such as effective security features and the emergency lines and services, can help to reduce the perceived risks in using Airbnb. The growth of the sharing economy, specifically Airbnb, is generating competition and challenges for the hotel/motel industry. The Airbnb hosts need to keep up with the professionalism in generating unique experiences and values for their consumers, which impacts customers' repurchase intention [133]. As such, our results also contribute to future promotional strategies across Airbnb activity service providers as they can be used as advertising content to reduce potential consumers' scepticism and enforce the benefits of the service.

*5.3. Limitations and Future Research*

The findings of the study should be used with caution considering that a convenience sample was used, which means that results cannot be generalized. The sample size is

another limitation. Future research with a larger sample size, done as a case study, is proposed to develop comparative studies. Future studies should involve probabilistic samples to provide representative results. Longitudinal studies should also be considered as they provide an overview of sequence of events overtime.

**Author Contributions:** Conceptualization, A.M.; methodology, J.L.; discussion, A.M.; results description, J.L., conclusion and implications, A.M. All authors have read and agreed to the published version of the manuscript.

**Funding:** There was no funding involved for this work.

**Data Availability Statement:** Data is only available to authors; it was not reported publicly.

**Conflicts of Interest:** There is no conflict of interest involved.

## Appendix A. Sample Characteristics

| | | | Type | | Total |
| | | | Non-Consumers | Consumers | |
|---|---|---|---|---|---|
| Gender | Female | Count<br>% within Gender | 68<br>51.9% | 63<br>48.1% | 131<br>100.0% |
| | Male | Count<br>% within Gender | 35<br>46.7% | 40<br>53.3% | 75<br>100.0% |
| Marital Status | Single | Count<br>% within Marital status | 57<br>52.8% | 51<br>47.2% | 108<br>100.0% |
| | Married | Count<br>% within Marital status | 46<br>46.9% | 52<br>53.1% | 98<br>100.0% |
| Children | 0 | Count<br>% within Children | 73<br>45.3% | 88<br>54.7% | 161<br>100.0% |
| | 1 | Count<br>% within Children | 18<br>64.3% | 10<br>35.7% | 28<br>100.0% |
| | 2 | Count<br>% within Children | 10<br>76.9% | 3<br>23.1% | 13<br>100.0% |
| | 3 | Count<br>% within Children | 2<br>100.0% | 0<br>0.0% | 2<br>100.0% |
| | 4 | Count<br>% within Children | 0<br>0.0% | 2<br>100.0% | 2<br>100.0% |
| Income | Below Average | Count<br>% within Income | 36<br>62.1% | 22<br>37.9% | 58<br>100.0% |
| | Around Average | Count<br>% within Income | 25<br>50.0% | 25<br>50.0% | 50<br>100.0% |
| | Above Average | Count<br>% within Income | 26<br>49.1% | 27<br>50.9% | 53<br>100.0% |
| | Not sure | Count<br>% within Income | 16<br>35.6% | 29<br>64.4% | 45<br>100.0% |
| Total | | Count<br>% within Income | 103<br>50.0% | 103<br>50.0% | 206<br>100.0% |

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
