# Peer review of "Airbnb Hospitality: Exploring Users and Non-Users’ Perceptions and Intentions"

_sustainability, doi:10.3390/su131910884_

Round 1

Reviewer 1 Report

The article is written in an appropriate way, but I think that the first person should be avoided.

In abstract, the objective is not identified. It is missing findings, conclusions and implications. It is necessary to make these aspects concrete.

In the introduction, the structure of the paper must be referred.

Describe how hypotheses are going to be tested in methodology.

“3.1 PLS Path Modelling and Multi-Group Analysis”

Is the title of this chapter well formulated? If yes, the second part is not explained.

Doubts in table 1: variables loadings, these values come from the literature or come from this research. This information is confused. What is the source?

There are some typographical typos.

References and citations are not according to the rules.

This paper is interesting for the readership of this journal. This research provides advance in the current knowledge.

Author Response

Thank you for your comments, please note:

  • All first person is removed in the manuscript. 
  • The objective is highlighted in the abstract, with findings and implications. 
  • Introduction now includes brief about testing of hypotheses. 
  • Clarity is provided in table-1 by amending the title in  last column.  The variables are adapted from literature as highlighted in the table-1, results are from analysis.   
  • We hope reference formatting will be autocorrected in production.  

Reviewer 2 Report

The article analyzes an interesting topic, anyhow I do not find the internal logic of the topic in relation to the selection of the journal (from the point of view of its focus). From the point of view of formal comments, I recommend that the authors list the cited sources in accordance with the journal template.

Author Response

The paper has now highlighted at different places where sustainability is addressed with its focus on sharing economy - Airbnb.  

We hope all reference formatting will be corrected during production process. 

Reviewer 3 Report

After reading the paper my general opinion is fairly positive, although I have some concerns and questions related to the model quality and sub-models comparability. To begin with, literature review is very comprehensive and actually nothing more could be added there. My only comment is related to the introduction, where Authors mention ‘experienced and inexperienced consumers of Airbnb’ (line 71) and then immediately move to the comparison between those who ‘have used or not used the platform’ (and then ‘consumers and non-consumers’). Although I understand the intention, I believe that these are three distinctive states – therefore, for the sake of clarity, I would recommend avoiding mentioning ‘inexperienced consumer’ as that would suggest consumers who already have some, limited experience with the platform, which is something different to ‘non-consumer’.

Hypotheses are well developed and supported by literature and the model is also adequate developed. Variables are properly assigned, but I believe it would be advisable to use more than two manifest variables / indicators per latent variable – first of all, I am not sure if computing Cronbach’s Alpha is valid for only two variables (and is there any point in doing it), secondly, it would be beneficial to be able to use confirmatory tetrad analysis in order to test, whether the relationship between manifest variables and latent variable is formative and reflective (and what approach was used in the presented research?)

My main concern is related to R2 values – some of them are fairly low, even for social science, and claiming that 0.281 is acceptable is a very bold statement – such model explains only a fraction of dependent variable’s variance – here slightly more than 70% of variance is left unexplained and actually that means that important, considerable variables and relationships exists, AND they are not included in the model. I believe authors provide some discussion on that, and explain consequences for the research.

And that leads to the next question – is it possible / valid /  rational to compare two models, one with fairly decent fit (R2s>0.5) and second with rather poor fit (R2s<0.35)? Certainly, statistical methods will provide differences and test their significance, but now we compare something which is quite correct, in terms of modelling the reality, with something which is rather incorrect – what consequences will that situation have for the quality of the results?

Authors also should note one important thing which may be observed in the model development – it was possible to obtain a valid, fitting model in a sample of users with some experience with the platform, and the model for the group of non-users is not fitted well. Perhaps actually those two groups would have two different, distinctive mechanisms of satisfaction creation, and therefore models that would represent them? Please note that it should be expected, that users and non-users will have different expectations, may perceive different categories of risks and particular risk areas etc., therefore maybe it is not a good idea to search for one model for both groups. If Authors consider that idea valid, it could be mentioned.

I am not sure if the last research question (‘have similarities in their behaviour’) was answered in the text, as the research was related rather to cognitive states (expectation, risk perception) than to different behaviours.

Author Response

We have carefully read your comments, thank you for making those observations. Following is submitted as our explanation:

  • Thank you very much for bringing this to our attention. We calculated our latent factors using the scales provided by the literature. Some of the variables had to be excluded due to low loadings. Cronbach’s Alpha measures the internal reliability of a multi-indicator variable, so it is applicable to a two-item factor. We have also used other reliability measures, such as composite reliability and average variance extracted (AVE) in light with the literature. CTA was not calculated as we followed previously validated scales. 
  • Thank you for your comment. We have used the cut-off point established in the literature (Falk and Miller, 1992) to assess our R2 results. All R2 are within the threshold established by the literature. 
  • As all models have R2 within the suggested threshold established in the literature, we assessed the differences between path coefficients in our models. The results of inter-model path coefficient comparison are presented in Table 5. 

  • References:

    Falk, R. F., & Miller, N. (1992). A primer for soft modelling. Akron: University of Akron Press.

Round 2

Reviewer 2 Report

I'm sorry, but I have to insist on my original position. I do not see the overlap of the topic and the main thematic axis of the journal, within which the authors submit their study. The formal side of the modified study is unacceptable, the contribution is not submitted to the review procedure in accordance with the rules of the journal.

Author Response

In response to the comment from the reviewer that the overlap of the topic and the main thematic axis of the journal is not noticed, we would like to state that the paper draws upon the special issue information and aligns it hospitality management and marketing. The paper within the context of the main aim of this special issue is to explore ‘real life’ problems and challenges linked to hospitality management and marketing and to find solutions to survive in the current global environment, has addressed the following:    

  • Contemporary trends in management and marketing of Airbnb hospitality service industries.
  • Survival of Airbnb require rethinking and realigning  as a hospitality business. 
  • The paper suggests new conceptual model and hypotheses necessary to analyze and evaluate current situation, establish future trends and help Airbnb hospitality businesses to meet the diversifying demand of hospitality consumers.
  • The paper cultivates an awareness of rapidly changing and intensifying competition to survive in the industry. It also helps to understand strategies and theories associated with hospitality service ethics, business sustainability and environment responsibility.

The paper is within the main aim of this special issue linked to hospitality management and marketing and to find solutions to survive in the current global environment. 

We believe like any other good journal the formatting aspect will be covered during the production process of the journal.   

Reviewer 3 Report

I read the updated text and your replies and unfortunately I found them rather unsatisfactory. First of all, nearly no adjustments were made to the manuscript, and those that were made are rather related to the editing, not to the content of the text.

When it comes to your replies, I expected at least providing some explanations or clarifications for the reader of the manuscript. Nevertheless:

  1. Calculating Cronbach’s alpha is rather debatable for two-item scale, and the fact that it measures the internal reliability does lead to the possibility to apply it to two-item scale. Please take a look:

Eisinga, R., Grotenhuis, M.t. & Pelzer, B. The reliability of a two-item scale: Pearson, Cronbach, or Spearman-Brown?. Int J Public Health 58, 637–642 (2013). https://doi.org/10.1007/s00038-012-0416-3

Anyway, since in that scenario CA has a tendency to underestimate the reliability it could be acceptable but still, readers deserved some explanation and discussion.

  1. Regarding to your second and third comment – please note that there are many other sources within the area of social / consumer research, which deem other, higher values of R2 to be acceptable:

Hair, J. F., Ringle, C. M., & Sarstedt, M. (2011). PLS-SEM: Indeed a silver bullet. Journal of Marketing theory and Practice, 19(2), 139-152.

Chin, W. W. (1998). The partial least squares approach to structural equation modeling. Modern methods for business research, 295(2), 295-336.

Please also note, that Falk and Miller proposed that value as a guideline, based only on experience (p. 78). Besides, there is one more, fairly important (#1) rule there – three indices for each latent variables. Why you did not follow this one?

  1. My comment, however, was not related to the values of R2 but to the possibility to compare models with fairly different fractions of endogenous constructs’ variances explained by their predictor constructs – how does it affect comparability?
  1. CTA does not have anything in common with scales being previously validated or not. Please refer to: Bollen, K. A., and Ting, K.-f. (2000). A Tetrad Test for Causal Indicators, Psychological Methods, 5(1): 3-22.

Finally, I do not consider your paper completely erroneous or faulty, since research in social science is fairly troublesome and problematic on many levels, esp. considering lack of reliability of the data gathered through questionnaire research, therefore, many approaches and points of view may be adopted and expressed, and it is not easy to obtain clear, interpretable results. Nevertheless, all these assumptions and nuances have to be properly documented and discussed.
